# Knowledge Discovery from Medical Data and Development of an Expert System in Immunology

**DOI:** 10.3390/e23060695

**Published:** 2021-05-31

**Authors:** Małgorzata Pac, Irina Mikutskaya, Jan Mulawka

**Affiliations:** 1The Department of Immunology, The Children’s Memorial Health Institute, Al. Dzieci Polskich 20, 04-730 Warsaw, Poland; m.pac@ipczd.pl; 2Institute of Computer Science, Warsaw University of Technology, Nowowiejska 15/19, 00-665 Warsaw, Poland; ira_stan@mail.ru

**Keywords:** artificial intelligence, expert systems, immunology, Bruton’s disease, CLIPS, data mining

## Abstract

Artificial intelligence is one of the fastest-developing areas of science that covers a remarkably wide range of problems to be solved. It has found practical application in many areas of human activity, also in medicine. One of the directions of cooperation between computer science and medicine is to assist in diagnosing and proposing treatment methods with the use of IT tools. This study is the result of collaboration with the Children’s Memorial Health Institute in Warsaw, from where a database containing information about patients suffering from Bruton’s disease was made available. This is a rare disorder, difficult to detect in the first months of life. It is estimated that one in 70,000 to 90,000 children will develop Bruton’s disease. But even these few cases need detailed attention from doctors. Based on the data contained in the database, data mining was performed. During this process, knowledge was discovered that was presented in a way that is understandable to the user, in the form of decision trees. The best models obtained were used for the implementation of expert systems. Based on the data introduced by the user, the system conducts expertise and determines the severity of the course of the disease or the severity of the mutation. The CLIPS language was used for developing the expert system. Then, using this language, software was developed producing six expert systems. In the next step, experimental verification was performed, which confirmed the correctness of the developed systems.

## 1. Introduction

### 1.1. The Need for Using Expert Systems in Medicine

An increase in the mortality of patients affected by various diseases can be observed in most developing countries. Among the reasons may be the lack of possibility to have a highly specialized consultation and insufficient number of specialist doctors, resulting in a delayed diagnosis and therapy. Low awareness of some rare diseases and lack of experience pertaining to their scope also result in delays in terms of their diagnosis and treatment [1]. Patients often waste their valuable time waiting for a doctor’s appointment. In this case, time works against the patient. Early treatment, even before the occurrence of complications, improves prognosis. The use of information technology may shorten the waiting time for an appointment with a specialist. Computer programs or applications based on artificial intelligence are helpful for doctors in decision-making without direct consultation with specialists [2]. Artificial intelligence (AI) is not intended to replace a specialist or doctor. The use of computer techniques is intended to help in diagnosing and proposing treatment methods [3]. It enables detection of dependencies in huge medical databases, which are subsequently used for treating and predicting the patient’s status in many clinical settings. These programs are designed to support healthcare professionals in their daily duties by helping them with the tasks of manipulating medical data and knowledge. A patient who has been diagnosed with high-risk symptoms may be shortlisted to continue treatment in a specialist. The use of innovative technologies, particularly AI techniques in medical applications, can reduce costs, time and medical errors. Its advantage is also the fact that AI does not omit any details and can be more accurate than a human doctor in terms of management of a given case [4]. The use of artificial intelligence in medicine has proved to be useful in detecting data patterns and is used in various types of experiments and clinical research to facilitate decision-making in each stage of diagnosis and subsequent treatment. Intelligent medical systems are developed to improve health care and provide services of better quality. The introduced systems support users by providing early diagnosis, treatment, as well as by predicting potential complications. Although the system uses “human” knowledge, it will never replace a specialist. Knowledge bases must be updated from time to time and, above all, controlled by human. Human being is an essential element for ensuring proper functioning of the whole system [2].

However an important role play expert systems [3,5,6,7,8,9,10] which are computer systems emulating the decision-making ability of a human expert. Expert systems are designed to solve complex problems by reasoning through bodies of knowledge, represented mainly as if–then rules rather than through conventional procedural code.

### 1.2. Short Consideration of Medical Expert Systems

Medical expert systems are computer programs that assist doctors in evaluating, diagnosing and treating patients. Medical expert systems are a type of artificial intelligence accessed through computer software that helps medical practitioners, such as hospital doctors, nurses and general practitioners, make informed decisions about patient care. Exemplary known medical expert systems are provided below.

MYCIN [11] was an early backward chaining expert system that used artificial intelligence to identify bacteria causing severe infections, such as bacteremia and meningitis, and to recommend antibiotics, with the dosage adjusted for patient’s body weight. The Mycin system was also used for the diagnosis of blood clotting diseases.

Caduceus [12] was intended to improve on MYCIN to focus on more comprehensive issues than a narrow field like blood poisoning, instead embracing all internal medicine. CADUCEUS eventually could diagnose up to 1000 different diseases.

Aidoc [13] evelops advanced healthcare-grade AI based decision support software. The program analyzes medical imaging to provide one of the most comprehensive solutions for flagging acute abnormalities across the body. It is helpful for radiologists.

DXplain [14] is a clinical decision support system available through the web that assists clinician by generating stratified diagnoses based on user input of patient signs and symptoms, laboratory results, and other clinical findings. The system also serves as a clinician reference with a searchable database of diseases and clinical manifestations.

Global Infectious Diseases and Epidemiology Online Network (GIDEON) [15] is a web-based program for decision support in the fields of infectious diseases and geographic medicine. Due to the advancement of both disease research and digital media, print media can no longer follow the dynamics of outbreaks and epidemics as they emerge in real time. In the program more than 300 generic infectious diseases occur and are challenged by over 250 drugs and vaccines. 1500 species of pathogenic bacteria, viruses, parasites and fungi have been described.

CASNET/Glaucoma [16] program deals with the diagnosis and treatment of eye diseases. It is implemented using the theory of representation of causal knowledge. The medical knowledge in Glaucoma is represented as a network of physiological states.

### 1.3. Aim and Scope of the Article

Modern medicine involves high hopes for the development of information technologies. The chronic shortage of doctors results in the development of new applications that will accelerate the development of modern diagnostics and medical therapy. New medical standards are expected. This article concerns these above-mentioned issues. This article includes cooperation with doctors of the Department of Immunology the Children’s Memorial Health Institute (CMHI) and concerns patients with Bruton’s disease. The main aim of the article is to discover knowledge from the obtained base and then establish an expert system based on this knowledge. Discovering of the most interesting decision trees that would enable detection of dependencies in groups of sick children with various mutations and different disease severity is one of objectives of the conducted analysis.

## 2. Brief Description of the Immunological Basis of Bruton’s Disease

Immunology is known to be a scientific discipline that examines our immunity, in particular how organisms react to pathogens. Immunology analyzes how the body’s defense mechanisms react against external factors, such as pathogenic microorganisms, toxins, etc. [17]. On the other hand, clinical immunology concerns the diagnosis and treatment of immunodeficiencies using various diagnostic and therapeutic tools. Molecular diagnostics and tests, including flow cytometric tests, immunophenotyping of peripheral blood and bone marrow lymphocytes, are conducted in diagnostic centers [17]. The Department of Immunology performs diagnostics of all types of primary immunodeficiencies, as well as their treatment, including immunoglobulin therapy and treatment with biopharmaceuticals [18]. X-linked agammaglobulinemia (XLA), known also as Bruton’s disease, is one of diseases that belong to the group of primary immunodeficiencies in which the Department of Immunology, CMHI specializes. The disease is the result of a mutation of Bruton’s tyrosine kinase (Btk), the gene located on the long arm of the X chromosome (Xq21.3-q22).

The maturity block occurs at the stage of differentiation of pro-B lymphocytes into pre-B lymphocytes, which means that immature B lymphocytes do not leave the marrow. The result is a very low percentage and number of B lymphocytes, as well as lack of production of immune antibodies (immunoglobulins). Susceptibility to various types of infections increases in an organism without immunoglobulins [9]. Boys with XLA usually begin to suffer from the disease after the age of 6 months. Before that age, the antibodies received from the mother via the transplacental route play a protective role. Recurring infections affect ears, nose, conjunctivitis, sinusitis and pneumonia; sometimes they may be subject to generalization. Gastrointestinal infections caused by Giardia lamblia are frequently observed. They cause abdominal pain, diarrhea, weight loss or growth inhibition. Infections may also affect bones, joints and skin. Despite recurring infections, the physical examination shows very small palatine tonsils and normal lymph nodes in patients.

The criteria for disease severity include early onset of signs and symptoms (infections), severe invasive infections: blood-borne pneumonia, cerebrospinal meningitis and/or encephalitis, sepsis, bone inflammation. According to observations of other authors, some mutations are considered to be linked to a more severe course of disease (frameshift mutation, nonsense mutation) or a less severe course of disease (splice site mutation, missense mutation) [18,19,20,21,22,23,24,25,26]. CMHI and other immunological centers provide diagnosis and treatment for this disease in children from all over Poland. In accordance with various reports, Bruton’s agammaglobulinemia occurs with a frequency of 1: 100,000 male newborns [17,18,21].

## 3. Discussion Concerning CRAN Package Repositories

The CRAN repository creates a centralized memory area in which various AI packages are stored. These objects are located there along with detailed annotations, code fragments and application examples. Nowadays, the official CRAN repository contains over ten thousand published packages. Furthermore, many other packages are available on the Internet. Most of these packages are categorized by task, which makes it possible to browse packages that follow certain themes [5]. The following packages used in this article for exploring data:

Rpart—a package used for determining decision trees

rpart.plot—a package used for visualization of decision trees

randomForest—a package used for analyzing and building models based on random forests

ggplot2—a visualization package

ggpubr—a visualization package

plyr—a data manipulation package

Hmisc—an analysis and visualization package

PerformanceAnalutics—a package used for conducting an analysis

e1071—a package used for creating models.

## 4. The Database Used

### 4.1. Description of the Exploratory Database

The database used in this article for discovering knowledge was an xls spreadsheet in its original form. The file contained one sheet in which the data of children suffering from a rare disease—Bruton’s disease (XLA)—were stored. Relevant parameters of the immune system were extracted from medical charts of 51 XLA patients out of almost 1700 patients with different inborn errors of immunity diagnosed at the Children’s Memorial Health Institute. Since the disease is very rare and analysis concerns retrospective data an inclusion of control group does not apply in this case. Otherwise all discussed parameters are within reference ranges for age and have no value to create a decision trees for XLA. Each patient was described with 37 attributes. Some attributes had commentary on the test results. The patient’s age was measured simultaneously in years and months. It was necessary to transform the base into a form that would enable an analysis. Some cells contained information that was not relevant to the analysis. The database also contained attributes that—due to their irrelevance—would not be used for discovering knowledge. Below we show all the columns that were included in the spreadsheet together with their description:

Age of onset—the age of the onset of first signs and symptoms, a numerical value with commentary in months or years

Age of dg—the age of diagnosis, a numerical value with commentary in months or years

IgG—immunoglobulin G level, a numerical value with verbal commentary (g/L)

IgA—immunoglobulin A level, a numerical value (g/L)

IgM—immunoglobulin M level, a numerical value (g/L)

B cells—B-lymphocyte level, a numerical value (%, k/μL)

Btk monocytes—Btk expression of monocytes, a numerical value (%)

Btk LiB—Btk expression of B lymphocytes, a numerical value (%)

Family history—cases of similar diseases that can be found in the family history

Sepsis—whether the patient suffered from disease concerned—‘yes’ if they suffered from the disease, an empty cell if not

Meningitis—whether the patient suffered from disease concerned—‘yes’ if they suffered from the disease, an empty cell if not

Pneumonia—whether the patient suffered from disease concerned—‘yes’ if they suffered from the disease, an empty cell if not

Bronchitis—whether the patient suffered from disease concerned—‘yes’ if they suffered from the disease, an empty cell if not

Otitis—whether the patient suffered from disease concerned—‘yes’ if they suffered from the disease, an empty cell if not

Diarrhea—whether the patient suffered from disease concerned—‘yes’ if they suffered from the disease, an empty cell if not

URTI—whether the patient suffered from disease concerned (upper respiratory tract infection)—‘yes’ if they suffered from the disease, an empty cell if not

UTI—whether the patient suffered from disease concerned (urinary tract infection)—‘yes’ if they suffered from the disease, an empty cell if not

Sinusitis—whether the patient suffered from disease concerned—‘yes’ if they suffered from the disease, an empty cell if not

Bronchiectasis—whether the patient suffered from disease concerned—‘yes’ if they suffered from the disease, an empty cell if not

Abscess—whether the patient suffered from disease concerned—‘yes’ if they suffered from the disease, an empty cell if not

Hemophilia A—whether the patient suffered from disease concerned—‘yes’ if they suffered from the disease, an empty cell if not

Ascariasis—whether the patient suffered from disease concerned—‘yes’ if they suffered from the disease, an empty cell if not

Conjunctivitis—whether the patient suffered from disease concerned—‘yes’ if they suffered from the disease, an empty cell if not

Arthritis—whether the patient suffered from disease concerned—‘yes’ if they suffered from the disease, an empty cell if not

Stomatitis—whether the patient suffered from disease concerned—‘yes’ if they suffered from the disease, an empty cell if not

Furunculosis—whether the patient suffered from disease concerned—‘yes’ if they suffered from the disease, an empty cell if not

Encephalitis—whether the patient suffered from disease concerned—‘yes’ if they suffered from the disease, an empty cell if not

Guillain-Barré—whether the patient suffered from disease concerned—‘yes’ if they suffered from the disease, an empty cell if not

Laryngitis—whether the patient suffered from disease concerned—‘yes’ if they suffered from the disease, an empty cell if not

IBD—whether the patient suffered from disease concerned (inflammatory bowel disease)—‘yes’ if they suffered from the disease, an empty cell if not

Mutation—the severity of a mutation detected regarded as:—severe, less severe

XLA—the severity of XLA disease—very severe, severe, less severe

Exon—numbers of exons in which changes were detected

Nucleotide change—changes in nucleotides due to mutations, verbal description

Protein change—changes in proteins due to mutations, verbal description

Other—other comments, verbal description

n/a—a numerical value in one line only

X-linked agammaglobulinemia as well as primary immunodeficiencies belong to rare diseases. Variables applied in the exploratory database are typically used in clinical practice. The group of 51 patients is really numerous. It should be mentioned we carried out research on statistical properties of Bruton’s disease. However it is a large material and it exceeds the frames of this article.

### 4.2. Database after Initial Modification

After consultation with an expert, the following changes were made: comments were removed from numerical columns, the age of the onset of first signs and symptoms and the age of diagnosis were changed from years and months to months only. After discussion with the expert it was decided to add measurable columns to the database:

Diseases were grouped by scope—an added column contained a number of transferred diseases from a given area (system, organ, e.g., rheumatology)

A column that showed the number of transmitted diseases of very severe course was added

A column that showed the number of transmitted diseases of less severe course was added

As the scope of reference values for IgG, IgA, IgM and B lymphocytes varies with age, these columns were transformed to represent the immunoglobulin level as a percentage of the standard value.

Two important columns with missing values were also included in the database. Considering that those columns were necessary for further analysis, additions were applied to them. When pre-processing, the number of attributes decreased significantly. The following attributes after pre-processing included:

Age ob—the age of the child in whom first signs and symptoms occurred

Age dg—the age of diagnosis

IgG%—immunoglobulin G level compared to the standard

IgM%—immunoglobulin M level compared to the standard

IgA%—immunoglobulin A level compared to the standard

Bcells Percentage—B-lymphocyte level compared to the standard

Btk LiB—Btk expression in B lymphocytes

Btk monocytes—Btk expression in monocytes

Family—cases of similar diseases and deaths resulted from them, which can be found in the family history

Severe diseases—the number of severe diseases transmitted

Less severe diseases—the number of less severe diseases transmitted

Rheumatology—the number of diseases transmitted in the area of

Pulmonology—the number of diseases transmitted in the area of

Otorhinolaryngology—the number of diseases transmitted in the area of

Gastroenterology—the number of diseases transmitted in the area of

Resistance—the number of diseases transmitted in the area of

Mutation severity—the severity of a mutation

Course—the severity of XLA disease.

The obtained columns will be used for building models based on which the samples will be classified into two types of classes: (a) due to mutation severity (severe, less severe), (b) due to disease severity (very severe, severe, less severe).

It should be emphasized that not obvious genotype-phenotype correlation has been established. Furthermore, no clear correlation between course of the disease and type of mutation or BTK expression on either B cells or monocytes has been found so far. Some mutations are regarded as a severe, some—as a less severe.

## 5. Exploring of the Database Held and Discussion

Decision trees, which are easy to interpret, are one of the most frequently used techniques in data analysis and knowledge discovery. This method works well when it comes to decision-making problems concerning multiple branching courses of action or making high-risk decisions [27]. Therefore, decision trees are frequently used in medicine, which facilitates the decision-making process regarding the physician’s work. The aim of discovering knowledge in a similar way is to find the best object splitting that will ensure maximum class homogeneity. In decision trees, using specific rules, a set of objects is split into subsets with similar features. Tree nodes contain splitting criteria, leaf nodes (leaves) include elements with an assigned label. Objects that can be found in one leaf node are similar to each other.

The aim of the article is to create a decision trees for Bruton’s agammaglobulinemia based on available retrospective clinical data, including the onset of clinical signs and symptoms, organ/systemic manifestation of signs and symptoms, levels of primary immunoglobulin classes (G,A,M), number of B lymphocytes, Btk expression in B lymphocytes and monocytes, type of mutation.

The analysis concerned retrospective data from patients with definitive diagnosis of XLA, and inclusion of a control group of healthy children did not apply in this case. Otherwise discussed parameters are within the reference ranges for age and cannot in any way create a decision trees for XLA.

Using the decision trees, tests were conducted to find features that could affect disease severity and severity of mutation. The leaves of obtained trees will contain subsets of objects split by the above-mentioned attributes. The sizes of classes in a set are given below.

The division by disease course: very severe—17, severe—22, less severe—12.

The division by mutation severity: severe—28, less severe—23.

The size of classes is quite similar. An analysis of the attribute importance was conducted using a random forest algorithm. Because random forests [28] belong to the category of methods for the construction of collective classifiers which utilize modifying of training data sets’ attributes. Random forest method utilizes decision trees solely as base classifiers. The essence of random forest lies in an additional variant reduction “collective classifier” by an additional reduction of mutual correlation between base classifiers. Random forests could be used for big data volumes with a great number of description attributes. The results of this analysis are shown in Table 1.

Based on the results obtained, it can be observed that the test results have the greatest importance in determining disease severity and severity of mutation. Past diseases and cases of similar diseases that can be found in the patient’s family history are of the least importance. The above-mentioned results will be proved in further steps of discovering knowledge in the process of building decision trees. To conduct the experiments, the data set was split into two subsets—training and testing. The testing set contains 70% of data, the training one—30%. The data sets should be representative, therefore, the createDataPartition function was used for even distribution among the testing/training sets. The rpart function was used for building decision tree models. The following parameters of a given function on the basis of which decision trees were built:

Formula—symbolic description of a model

Date—the data matrix from which attributes were used for building a model

Control—parameters used for controlling a tree structure

Method—a tree-building method (a class method used)

Minsplit—a minimum number of observations in the node used for splitting

Maxdepth—maximum tree-depth.

The decision trees of the greatest interest are set out below.

### 5.1. Decision Trees Generated to Determine Disease Severity

Based on an analysis of the attribute importance for the purposes of experiments, a model was built with the parameters described below.

The attributes used for building the model: IgG%, IgA%, IgM%, Age of the onset of signs and symptoms

Parameters that control the tree structure: Minsplit, Maxdepth.

A decision tree shown in Figure 1 was built using the above-mentioned parameters.

After the test data have been checked, the accuracy of the obtained model is equal to 0.79. This means a high value. The above graph shows that the immunoglobulin M level may have a large effect on disease severity. Subsequently, the Age of the onset of first signs and symptoms was an attribute that was used for splitting. An early onset of first signs and symptoms also determines the course of Bruton’s disease—the earlier first signs and symptoms, the more severe disease course in a patient. The results of immunoglobulin (A and G) tests were then used for the splitting. The lower the immunoglobulin level, the more severe disease course.

In the next step of tree generation, such attributes as the percent of B cells, Btk in B lymphocytes and Btk in monocytes were used. Parameters that control the tree structure include Minsplit—8, Maxdepth—5. The above-mentioned parameters have been selected so that the model gives the best possible result. The model accuracy using these parameters is 0.64. Visualization of a decision tree is shown in Figure 2.

The built model shows quite a good result. The aforementioned tree shows that disease severity depends on B lymphocytes as well. A low level of Btk expression and the absence or very low level of B lymphocytes in patients’ blood may result in a severe course of Bruton’s disease. On the other hand, a higher Btk expression of B lymphocytes and a higher level of B cells in blood indicates less severe disease course.

The next stage of tree generation was conducted using such attributes as Rheumatology, Pulmonology, Otorhinolaryngology, Gastroenterology, Resistance. This experiment has investigated whether there is a relationship between the occurrence of individual diseases in patients and the severity of Bruton’s disease. The best result was obtained using such parameters as Minsplit—7, Maxdepth—5. The model accuracy was 0.36. This value is not high enough to apply the built model in practice.

In the next step of tree generation, the dependency of disease severity on such parameters as mutation severity, disease seriousness (severe infections), the number of less severe diseases transmitted, the occurrence of similar cases in the family history and the age of the onset of first signs and symptoms, was analyzed. The model obtained the best results using such parameters as Minsplit—5, Maxdepth—5.

In this case, the model accuracy was 0.93. This is a good result of the model, which means that disease severity is influenced by the attributes that have been used in that model. Figure 3 shows a decision tree used for the model concerned. Based on the model obtained, it can be stated with almost a hundred percent certainty that if the patient has been diagnosed with a severe mutation, has suffered from severe diseases and first signs and symptoms have been detected before the age of two, the course of Bruton’s disease is very severe. However, if the mutation is less severe and first signs and symptoms have been detected after the age of two, disease course is less severe.

### 5.2. Results Decision Trees Created for Identifying the Mutation Severity

The authors will now identify mutation severity depending on the test results and other signs and symptoms that occur in children with Bruton’s disease. As in the previous subsection, the first experiment made it possible to examine the dependency of mutation severity on the level of immunoglobulins A, G and M. The best result of the built model was obtained using such parameters as Minsplit—4, Maxdepth—5. The model accuracy concerning the test data reached 71%. This is a good result, which makes it possible to use the model concerned for identifying mutation severity based on the immunoglobulin level. Figure 4 shows a decision tree used for the model concerned.

When analyzing the above-mentioned tree it can be observed that a lower immunoglobulin level in a patient may result in a severe mutation. The immunoglobulin M level was the first parameter under analysis, which was used for splitting. At the undetectable level of IgM (43%), already in the first step of tree generation, the observations were attributed to severe mutations. Only one of them was misclassified. In the next step, the immunoglobulin G level was a decision attribute. Thirty percent of observations with a level higher than an undetectable one were attributed to a group of less severe mutations. In this step, only one observation was misclassified as well. The immunoglobulin A level is a third decision-making attribute. One hundred percent of observations with reduced IgA level and compliance with previous conditions were attributed to severe mutations. While respecting all the above-mentioned rules and lowering the IgM level, 8% of the observations were attributed to severe mutations, the others to less severe ones. There was one misclassified observation in each leaf node.

In the next step, the dependency of mutation severity on such parameters as Btk expression in B lymphocytes, Btk expression in monocytes, B-lymphocyte level in blood and the onset of first signs and symptoms of disease was analyzed. After several experiments with modification of parameters of the decision tree structure were conducted, the best result was obtained with such parameter values as Minsplit—4, Maxdepth—5. The model accuracy concerning the test data reached 86%. A tree diagram is shown in Figure 5.

The first parameter that determines mutation severity according to the above-mentioned tree is Btk expression in B lymphocytes. The expression higher than 20 indicates a less severe mutation. The next parameter used for splitting is the age of the onset of first signs and symptoms. If first signs and symptoms occur after the age of two and Btk expression in B lymphocytes is less than 20, the mutation is severe. The level of B lymphocytes in the patient’s blood is a third attribute that was used for splitting. The absence or a very low level of lymphocytes indicates a severe mutation.

The next tree which analyses the dependency of mutation severity on different parameters was based on the study of the dependencies on past severe and less severe diseases and on the occurrence of Bruton’s disease in the family history. The model achieved the best result using such parameters as Minsplit—4, Maxdepth—5. The model accuracy concerning the test data is 79%. A decision tree used for the model concerned is shown in Figure 6.

The illustrated tree shows that the first splitting criterion is the number of less severe diseases suffered by the patient. Such attributes as Pulmonology and Otorhinolaryngology were used for building the model. A larger number of past diseases may be evidence of a severe mutation.

## 6. Establishing an Expert System Used for Diagnostic Applications

### 6.1. The CLIPS Language Selection

Expert systems are one of the most widespread and applied systems in the field of artificial intelligence [3,6,7,8,9]. Those systems use the experience of specialists in areas in which the quality of decision-making depends on the level of expertise. The use of expert systems is only effective in specific expert areas in which the empirical experience of specialists is of great importance.

The language of expert systems—CLIPS (C Language Integrated Production System) [29]—has been used for implementing the knowledge base in this article. CLIPS was created in NASA. It is a forward-chaining rule-based programming language that also provides procedural and object-oriented programming facilities. CLIPS is high quality, excellent documentation, very impressive and regularly updated. It is one of the languages more frequently used to build complicated expert systems.

Building classic expert systems using this tool involves describing the system behavior by defining necessary rules [10]. Those rules describe actions to be performed in response to the fulfilment of certain conditions. Those conditions come from information that can be found in the working memory of the expert system. The expert system has a special inference mechanism which matches the facts that can be found in the working memory or on a fact list used for defining rules. This mechanism also determines which rules can be run.

The main program is written in the CLIPS language. It uses the knowledge obtained from decision trees that were previously generated. Individual decision trees can be saved as rule sets. Any decision tree could be easily transformed into a set of classification rules. A rule is generated for each decision tree path connecting the root of the tree with a leaf. To illustrate, one should consider a decision tree depicted in Figure 1. This tree could be transformed into subsequent set of seven rules.

The record of this tree is as follows:

IF {IgM% is undetectable and the age of the onset of first signs and symptoms is

early; IgG% is undetectable} THEN {Bruton’s disease course is extremely severe};

IF {IgM% is undetectable and the age of the onset of first signs and symptoms is

early; IgG% is detectable} THEN {Bruton’s disease course is severe};

IF {IgM% is undetectable and the age of the onset of first signs and symptoms

is late} THEN {Bruton’s disease course is severe};

IF {IgM% is detectable and the age of the onset of first signs and symptoms

is early; IgA% is undetectable} THEN {Bruton’s disease course is severe};

IF {IgM% is detectable and the age of the onset of first signs and symptoms

is early; IgA% is greater than 50} THEN {Bruton’s disease course is severe};

IF {IgM% is detectable and the age of the onset of first signs and symptoms

is early; IgA% is from the range (20, 50)} THEN {Bruton’s disease course is less severe};

IF {IgM% is detectable and the age of the onset of first signs and symptoms

is late} THEN {Bruton’s disease course is less severe}.

### 6.2. Brief Description of the Created Software

The expert system created comprises two parts: the interface in Java and the expert system itself in the CLIPS language. Software including approximately 600 lines of code was written to create the expert system. The created interface is used for communication and control of the interference process.

Its operation is as follows. After the application has been started, the user must select the option what the expertise will be about (identification of disease severity or mutation one). Based on the selected option the expertise will be conducted using three groups of enterable data. After the selection has been made, appropriate text fields will be highlighted, while others will not be activated. The values entered by the user in the active fields are verified at the same time. All entered values should be numeric (integer numbers). Subsequently, an expertise is carried out. Based on the facts, which can be found in the fact base and the rule-based knowledge base (implemented in the CLIPS language), the system makes forward-looking conclusions. Its result is returned to the user via the interface.

## 7. Verification of the Implemented Expert Systems

To verify the implemented expert systems 33 tests were performed. The experiments comprised of defining the severity of Bruton disease course and mutation by the use of the application. The system made decisions based on data prepared beforehand. To verify the expert systems test cases, which included every possible decision path in the system, were considered. The tests comprised of a comparison between results achieved by the system and those expected. Two examples of the experiments are shown below.

The tests were conducted to identify the severity of Bruton’s disease and mutation severity using the created application. The system is to make decisions based on previously prepared data. The presented test cases, which take into account each decision-making path in the established system, were used to verify the implemented expert systems. The tests are used for comparing the obtained results that were returned by the system together with the expected ones. The sets of test data were created based on parameter values taken from recent clinical measurements. These sets consist of age of onset and values of IgM, IgG and IgA.

**Test 1.** Verification of disease severity based on the results of immunoglobulin tests

Seven sets of test values were used to test a decision tree shown in Figure 1. The result of system operation for the first set of test data is shown in Table 2.

**Test 2.** Verification of mutation severity based on the results of immunoglobulin tests

Five sets of test data were created to verify a decision tree shown in Figure 4. They can be found together with the obtained results in Table 3.

The aim of the experiments was to verify the implemented expert systems based on decision trees obtained from the previously shown results. The verification process was successful. No errors were found in the inference process; it followed the decision-tree paths.

## 8. Summary

This article attempted to establish an expert system that would be helpful for doctors in identifying both the severity of mutation that occurs in patients with Bruton’s disease and the severity of this disease. By discovering knowledge using decision trees, 6 models were identified, which resulted in an interesting result that made it possible to apply them in an expert system. While working on the article, a prototype of an expert system was created. The CLIPS language was applied. The above-mentioned system enables identification of mutation severity in patients with Bruton’s disease and the severity of its course. According to the authors, the main achievements of this work include the knowledge obtained from the databases, which will be used for supporting clinical diagnosis using computer techniques in the early stages not only on the basis of conducted clinical trials, but also logical reasoning.

## Figures and Tables

**Figure 1 entropy-23-00695-f001:**
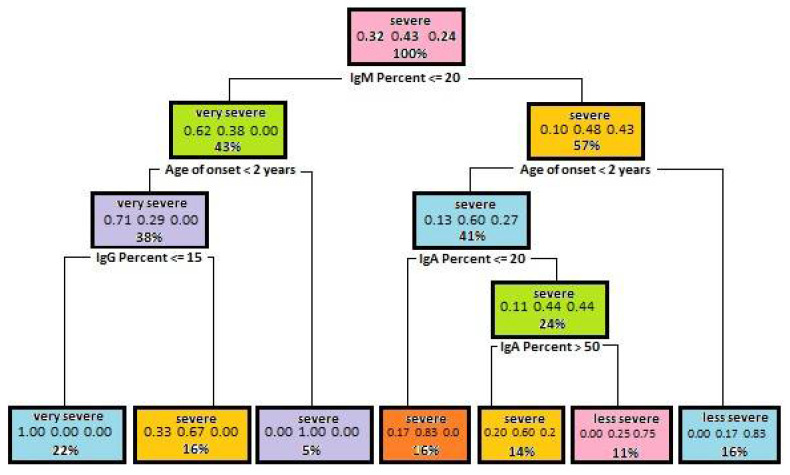
A dependency tree diagram, which illustrates dependencies of disease severity on the level of immunoglobulins and the age of the onset of first signs and symptoms of disease.

**Figure 2 entropy-23-00695-f002:**
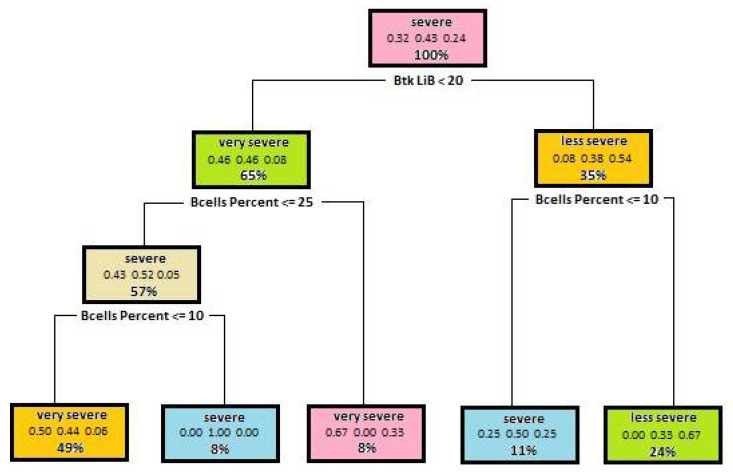
A dependency tree diagram, which illustrates dependencies of disease severity on Btk expression in B lymphocytes and the number of those lymphocytes in the patient’s blood.

**Figure 3 entropy-23-00695-f003:**
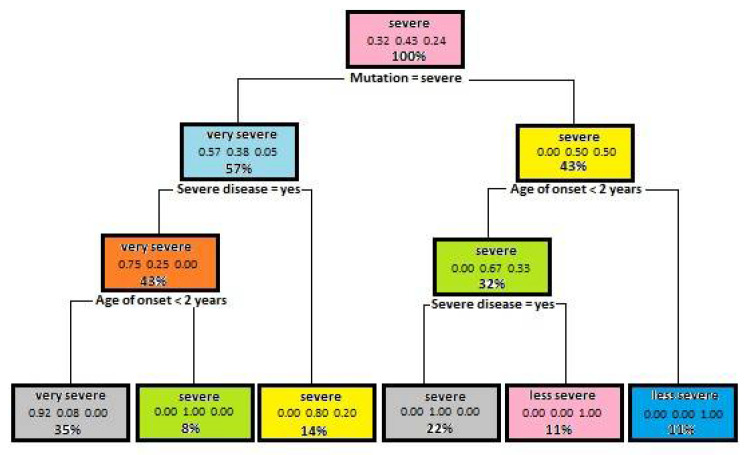
A dependency tree diagram, which illustrates dependencies of disease severity on selected parameters.

**Figure 4 entropy-23-00695-f004:**
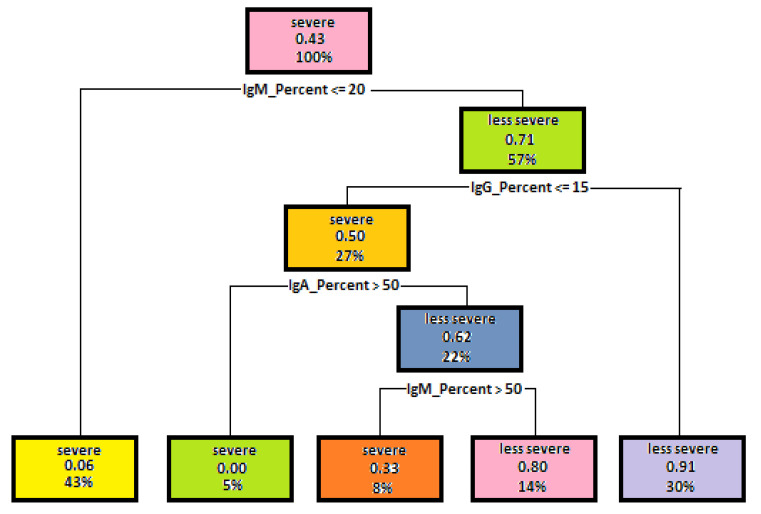
A dependency tree diagram, which illustrates dependencies of mutation severity on the immunoglobulin level.

**Figure 5 entropy-23-00695-f005:**
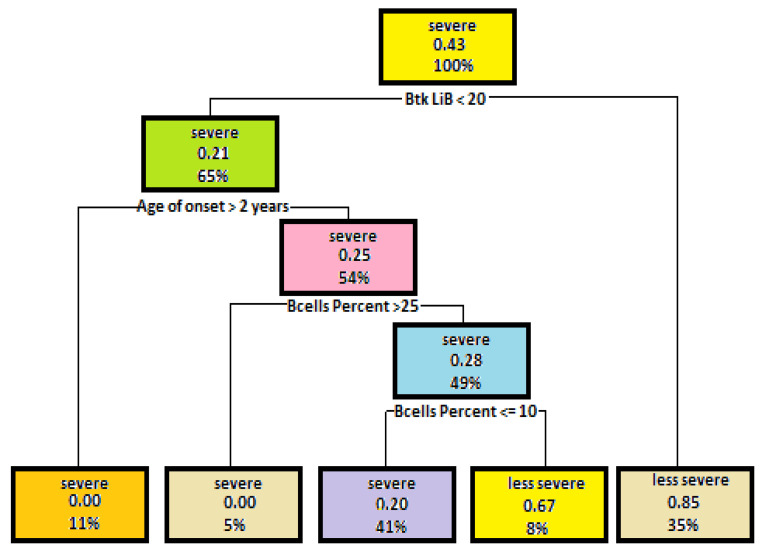
A dependency tree diagram, which illustrates dependencies of mutation severity on the test results of Btk expression in B lymphocytes, B-lymphocyte level and the age of the onset of first signs and symptoms of disease.

**Figure 6 entropy-23-00695-f006:**
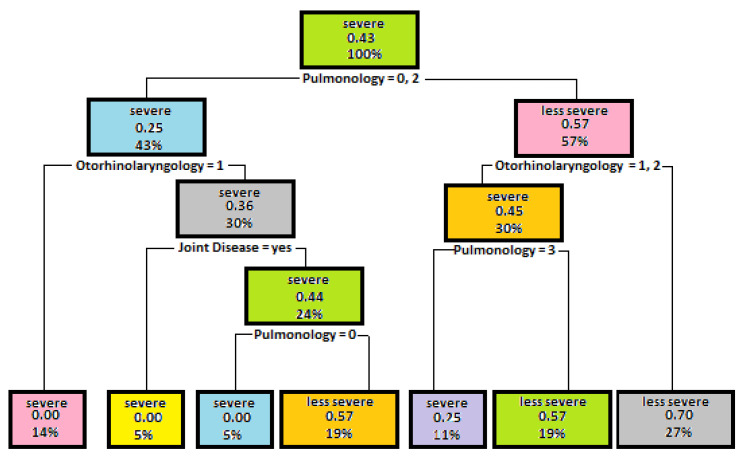
Presentation of the dependencies of mutation severity on more or less severe past diseases.

**Table 1 entropy-23-00695-t001:** List of the attribute importance by disease severity and mutation severity.

Attribute	Gini Index for Disease Severity	Gini Index for Mutation Severity
Age of onset	2.3768844	1.3391285
Percentage of IgG	3.1353790	3.5060282
Percentage of IgM	6.2144411	4.2951580
Percentage of IgA	3.1072242	2.0747686
Percentage of B cells	2.6674329	1.9705842
Btk in B cells	4.9642351	4.5973863
Btk in monocytes	2.4322656	2.1629096
Family history	0.2389207	0.2670715
Severe diseases	1.7318287	1.3331821
Less severe diseases	0.6482581	1.3391285
Rheumatology	0.2090056	0.2530046
Pulmonology	0.9603348	0.7524785
Otorhinolaryngology	0.3649750	0.5219833
Gastroenterology	0.4077892	0.4276909
Immunology	0.3829554	0.5736574
Mutation severity	2.8436096	---

**Table 2 entropy-23-00695-t002:** Test data sets used for testing a decision tree shown in Figure 1.

	Age of Onset	IgM	IgG	IgA	Expected Result
Data set 1	19	12	19	17	Very severe course of the disease
Data set 2	19	12	73	17	Severe course of the disease
Data set 3	47	12	73	17	Severe course of the disease
Data set 4	47	92	73	17	Less severe course of the disease
Data set 5	23	92	73	17	Severe course of the disease
Data set 6	23	92	73	46	Less severe course of the disease
Data set 7	23	92	73	76	Severe course of the disease

**Table 3 entropy-23-00695-t003:** Test data sets used for testing a decision tree shown in Figure 4.

	Age of Onset	IgM	IgG	IgA	Expected Result
Data set 1	23	17	73	29	Severe mutation
Data set 2	19	36	58	17	Less severe mutation
Data set 3	3	81	14	62	Severe mutation
Data set 4	120	81	7	35	Severe mutation
Data set 5	23	41	7	35	Less severe mutation

## Data Availability

All datasets generated for this study are included in the article.

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
