# Peer review of "Knowledge Discovery from Medical Data and Development of an Expert System in Immunology"

_entropy, 2021, doi:10.3390/e23060695_

Round 1

Reviewer 1 Report

This study presents the result of usage the database with information of patients suffering from Bruton’s disease employing data mining. The system proposed can conduct the expertise and determine the severity of the course of the disease or the severity of the mutation. The experimental analysis has been performed, and, in opinion of the authors, can confirm the correctness of the proposed expert system.

The authors used decision trees that can enable the detection of dependencies in groups of sick children with various mutations and different disease severity that is the objective of their analysis.

The proposed decision trees method for Bruton's agammaglobulinemia is based on clinical data, including the onset of clinical signs and symptoms, organ/systemic manifestation of signs and symptoms, levels of primary immunoglobulin  classes (G,A,M), number of B lymphocytes, Btk expression in B lymphocytes and monocytes, type of mutation.

 The idea was to select the features that could affect disease severity and severity of mutation. The leaves of obtained trees should contain the subsets of objects split by the above-mentioned attributes. The sizes of classes in a set are given: The division by disease course: very severe – 17, severe – 22, less severe – 12. The division by mutation severity: severe – 28, less severe – 23.

In experiments, the testing set contained 70% and training one - 30%. The proposed expert system has been implemented in the language of expert systems – CLIPS.

Two tests were realized to identify the severity of Bruton's disease and mutation severity using the created application: verification of disease severity based on the results of immunoglobulin tests; verification of mutation severity based on the results of immunoglobulin tests.

Only conclusion presented by authors was: “two tests are used for comparing the obtained results that were returned by the system together with the expected ones.” , that , in opinion of this reviewer is not sufficient for understanding od the performance of the designed expert system.

The authors presented the only conclusion: the verification process was successful; no errors were found in the inference process. They did not provide any another discussion or comments.

In summary, the authors wrote: “by discovering knowledge using decision trees, 6 models were identified, which resulted in an interesting result that made it possible to apply them in an expert system.” In opinion of this reviewer, the authors should explain more details of these six models into the text of manuscript.

 The authors should explain the values parameters that they used in Figs 1-3 that were presented in blocks of the diagrams, it is difficult understand for potential reader these values without additional explications.

 The authors did not provide any comparison with other variants of system for diagnostics used in practice. It is difficult understand the scientific contribution of this study without such comparison.

Author Response

Please find attached responses to reviewer one. 

Thank you. 

Reviewer 2 Report

GENERAL CONSIDERATIONS:
This is an interdisciplinary work between physicians from the Children's Memorial Health Institute in Warsaw and engineers from the Institute of Computer Science. The aim is to discover new knowledge about Bruton's disease by applying data mining techniques based on decision trees.

To the best of my knowledge, this is the first attempt to make a scientific contribution by applying artificial intelligence techniques to the study and classification of Bruton's disease. In itself it is a novel and interesting idea. However, I believe that the article requires significant improvements before being published in Entropy.

My comments and suggestions are intended to contribute ideas to the authors in order to show the full potential of the research they submit for scientific review.

Abstract:
The authors make a very general description that really should be done in the introduction. I suggest specifying more about the details from a technological perspective linked to data mining. Particularly decision trees and their main results.

1. Introduction:
The introduction explains the need to use expert systems in medicine and the objectives and scope of the article presented. A bibliographic review of similar expert systems applied to similar diseases is lacking. It should be contextualized about the options available within artificial intelligence. Particularly, it should be explained why decision trees are a good option compared to others such as neural networks, linear and logistic regression, etc. A new subsection is necessary about this issue.

1.1. The need for using expert systems in medicine:
References 3 and 5 are in Polish. For the sake of clarity, I suggest adding equivalent references in English.

2. Brief description of the immunological basis of Bruton's disease 73:
I consider this section to be well thought out. The need and scope of an expert system is correctly defined. In addition, the argumentation is understandable for non-expert professionals in immunology.

3. Discussion concerning CRAN package repositories:
This section is unnecessary. I suggest a brief description in other section, for example, 5.

4.1. Description of the exploratory database:
The number of patients is low. This is normal for such a rare disease. In any case, in addition to describing the variables that have been used for the study, it is necessary to justify why these variables have been used. Perhaps explain them by groups and relate them to a reasonable suspicion that they could have a discriminating capacity for the disease that is the subject of this article. On the other hand, in articles of this nature it is common practice to accompany the descriptions of the variables with statistics such as number of samples, mean, deviation, median, interquartiles, confidence intervals, p-values, etc., as appropriate according to the nature of each variable (dichotomous, categorical, numerical, etc.).

4.2. Database after initial modification:
Same considerations as in subsection 4.1.
Also, regarding this paragraph "The obtained columns will be used for building models based on which the samples will be classified into two types of classes: a) due to mutation severity (severe, less severe), 237 b) due to disease severity (very severe, severe, less severe).". It seems very important for the objectives of this paper. I suggest further justifying the rationale for these two classes.

5. Exploring of the database held and discussion:
References 16 (and 17 in section 6) are in Polish. For the sake of clarity, I suggest adding equivalent references in English. This section needs more references on decision trees, their use in medical applications, etc.

"An analysis of the attribute importance was con-262 ducted using a random forest algorithm.". Please, explain why.

The six models are constructed in an inductive and reasonable way, where the descriptive and classification potential of several variables is studied, and promissing accuracy results are obtained. However, more experiments and further maturing of the procedure are necessary to be able to draw scientifically relevant and credible conclusions.

"The testing set contains 70% of data, the training one – 30%." Given the low number of patients and data, and that it is very difficult to know the casuistry that may occur, repeated experiments should be performed. For example, by cross-validation, leave-one-out, or multiple repetitions (with or without resubstitution) with random selection of which patients/sample are used for training and testing. In this case the quality metrics could be given in the form of mean and variance, p-values, etc.

Accuracy is the only used quality metric for the classifier, but others such as sensitivity, specificity, recall, f1-score, etc. should also be provided in order to encrease the credibility of the results and conclusions.

5.2. Results Decision trees created for identifying the mutation severity:
Similar considerations as the ones made for section 5.1.

6. Establishing an expert system used for diagnostic applications:
6.1. The CLIPS language selection:
Impossible to rich the link in[18]: http://mblach-531 nik.pl/lib/exe/fetch.php/dydaktyka/zajęcia/se/lab12_clips.pdf ... Shouldit be http://www.clipsrules.net/?
It is not justified why the authors use CLIP instead of other programing environments like Python, Matlab, etc.

From lines 418 to 433 it seems an algorithm. The way in which the authors express the algorithm is unusual in the scientific-technical field. I suggest a more appropriate way to expres the algorithm.

7. Verification of the implemented expert systems:
It is clear the objectives of Test 1 and 2, but it is no clear how the sets of test data were created.

Author Response

Please find attached responses to reviewer two. 

Thank you. 

Round 2

Reviewer 1 Report

The authors have attended the comments

Author Response

Thank you